# Towards Blue AIE/AIEE: Synthesis and Applications in OLEDs of Tetra-/Triphenylethenyl Substituted 9,9-Dimethylacridine Derivatives

**DOI:** 10.3390/molecules25030445

**Published:** 2020-01-21

**Authors:** Monika Cekaviciute, Aina Petrauskaite, Sohrab Nasiri, Jurate Simokaitiene, Dmytro Volyniuk, Galyna Sych, Ruta Budreckiene, Juozas Vidas Grazulevicius

**Affiliations:** 1Department of Polymer Chemistry and Technology, Kaunas University of Technology, Radvilenu rd. 19, LT-50254 Kaunas, Lithuania; 2Department of Biochemistry, Lithuanian University of Health Sciences, A. Mickeviciaus st. 9, LT-44307 Kaunas, Lithuania

**Keywords:** tetra-/triphenylethene, acridan, aggregation induced emission enhancement, electroluminescence

## Abstract

Aiming to design blue fluorescent emitters with high photoluminescence quantum yields in solid-state, nitrogen-containing heteroaromatic 9,9-dimethylacridine was refined by tetraphenylethene and triphenylethene. Six tetra-/triphenylethene-substituted 9,9-dimethylacridines were synthesized by the Buchwald-Hartwig method with relatively high yields. Showing effects of substitution patterns, all emitters demonstrated high fluorescence quantum yields of 26–53% in non-doped films and 52–88% in doped films due to the aggregation induced/enhanced emission (AIE/AIEE) phenomena. In solid-state, the emitters emitted blue (451–481 nm) without doping and deep-blue (438–445 nm) with doping while greenish-yellow emission was detected for two compounds with additionally attached cyano-groups. The ionization potentials of the derivatives were found to be in the relatively wide range of 5.43–5.81 eV since cyano-groups were used in their design. Possible applications of the emitters were demonstrated in non-doped and doped organic light-emitting diodes with up to 2.3 % external quantum efficiencies for simple fluorescent devices. In the best case, deep-blue electroluminescence with chromaticity coordinates of (0.16, 0.10) was close to blue color standard (0.14, 0.08) of the National Television System Committee.

## 1. Introduction

Organic fluorophores emitting prompt blue fluorescence are required by industry since they are characterized by many advantages including high photoluminescence quantum yield (PLQY), fast fluorescent decays (in ns range, which are of interest not only for displays but also for visible light communications), good blue color purity, chromaticity coordinates which meet the National Television System Committee requirements, etc. [1,2,3,4]. Despite lower theoretical maximum of internal quantum efficiency (25%) of singlet emission based electroluminescent devices in comparison to that of phosphorescent [5] and thermally activated delayed fluorescence (TADF) [6] based devices, blue fluorescent emitters are used in commercial organic light-emitting diodes (OLEDs) of display and lighting technologies [7]. Such interest in blue fluorescence is mainly explained by its higher stability under electrical excitation, than that of mentioned blue phosphorescent and TADF emitters, the stability of which is fundamentally limited due to the presence of “hot” exciton forming long-lived triplet excitons [8,9]. Blue fluorescent emitters can be used for a novel energy-saving variety of OLEDs called hyper-fluorescent OLEDs, which are based on energy transfer from the TADF host to the fluorescent emitter in the light-emitting layer [10]. PLQY of OLED emitters (light-emitting layers) has to be close to 100%, which is not a simple task since aggregation-induced quenching is a common property of organic fluorophores [11,12,13]. To increase PLQY of emitters in solid state, one of the most promising molecular design strategies is introduction of moieties, prompting aggregation-induced emission (AIE) or aggregation induced emission enhancement (AIEE). Tetraphenylethene and triphenylethene are such moieties which usually give significant rise of PLQYs in solid state of many fluorophores including OLED emitters [14,15]. Notably, materials exhibiting AIE or AIEE phenomena are established as multifunctional materials. They are useful not only for OLEDs but also for chemical sensing, for detection of stimuli responses, bio and surface visualizations etc. [16].

Compounds based on carbazole [17,18,19] or triphenylamine [20,21] moieties substituted by tetraphenylethene and/or triphenylethene units were previously reported to show AIE or AIEE effects. Due to the specific linkage topology, some of them displayed blue/sky-blue fluorescence [22,23]. In addition to carbazole and triphenylamine, nitrogen-containing heteroaromatic acridan derivatives are also well established in OLED technology, some of them as blue emitters. It is expected that tetra-/triphenylethenyl substituted acridanes can exhibit high PLQYs in solid-state. We aimed to check this expectation in the current work. Deep-blue fluorescent OLEDs were recently developed achieving theoretical limit of EQEs (5%) [24]. Such high efficiencies of fluorescent OLEDs were explained by usage of novel emitters with *tert*-butyl substituents which inhibit dimer formation and crystallization-induced emission reduction. It was therefore of interest to use *tert*-butyl-substituted acridan moiety in the design of our emitters with potential ability of aggregation induced/enhanced emission. In the design of new materials for OLEDs, not only high luminescence efficiency but also proper HOMO and LUMO energy levels are of great importance to ensure efficient hole/electron injection under external electric fields [25]. Both acridan and tetra-/triphenylethenyl are electron-donating units and compounds containing these moieties are expected to have low ionization potentials. To control ionization potentials of acridan derivatives, electron-accepting cyano substituents were introduced. 

In this work, we report on the design and synthesis of new acridan-based emitters containing tetra-/triphenylethene units. Photophysical, thermal, electrochemical, and electroluminescent properties of the synthesized compounds were investigated to demonstrate the effect of different substitutions.

## 2. Results and Discussion

### 2.1. Synthesis

The synthetic route towards the targeted 9,9-dimethylacridine derivatives is shown in Scheme 1. Compounds **1**–**6** were synthesized by a one-step procedure, i.e., by the Buchwald-Hartwig method [26,27] in the presence of palladium complex. The synthesized compounds were identified by mass-, IR- and ^1^H, ^13^C NMR spectrometries.

### 2.2. Thermal Properties

The thermal characterization of the compounds was performed by DSC and TGA under nitrogen atmosphere (Figure 1). The thermal characteristics are summarized in Table 1. All the synthesized compounds (**1**–**6**) showed one-step thermal degradation with moderately high thermal stability (Figure 1b). The temperatures of five percent weight loss (T_d_) were in the range of 294–327 °C, as confirmed by TGA with the heating rate of 20 °C/min. T_d_ of compounds containing no *tert*-butyl groups (**1**–**3**) were found to be lower than those of the respective *tert-*butyl substituted compounds (**4**–**6**). Compounds **3** and **6** containing tetraphenylethenyl moieties exhibited higher T_d_ values than compounds **1** and **4** containing triphenylethenyl units. All the synthesized compounds could be used for the formation of the layers using a vacuum evaporation.

All the compounds were obtained as crystalline substances. Compounds **1**–**6** showed melting temperatures (T_m_) which in the range from 192 to 290 °C (Figure 1a, Appendix A). Melting points of *tert-*butyl substituted compounds (**4**–**6**) were found to be higher than those of the respective compounds without such substituents (**1**–**3**). Triphenylethenyl-substituted compounds **1** and **4** possess considerably lower melting points as compared to compounds substituted by tetraphenylethenyl units (**3** and **6**) [28]. Introduction of larger substituents such as cyano-group (compounds **2**,**5**) and phenyl (compounds **3**,**6**) fragments instead of hydrogen (compounds **1**,**4**) in the ethylene unit highly increased the T_m_ and T_d_ values of compounds (Table 1). Moreover, the introduction of bulky *tert*-butyl groups apparently affects the crystal packing between adjacent molecules and conformational arrangements of acridan units that resulted in increased T_m_ values for compounds **4**–**6**, with respect to non-substituted compounds **1**–**3** [29,30]. The attachment of *tert*-butyl substituents seems to make structures even more bulky and rigid which presumably leads to the enhanced T_m_ of compounds **4**–**6** as well as to their tendency to crystallize from liquid phase [30]. Compounds **4** and **5** also showed polymorphism (Figure 1a). Their samples exhibited two endothermal melting peaks in the first DSC heating scans (Table 1). When the melt samples of compounds **1**–**3** and **6** were cooled down during DSC experiments, they formed molecular glasses with glass transition temperatures (T_g_) in the range of 55–105 °C. Molecular glasses of compounds **2** and **6** were not morphologically stable; they tended to crystallize on further heating. Molecular glasses of compounds **1** and **3** did not show inclination to crystallization in DSC experiments. Compound **1** containing triphenylethenyl moiety showed considerably lower T_g_ (by 27 °C) than its analogous compound **3** containing tetraphenylethenyl species. Such observation can be explained by higher molecular weight and stronger intermolecular interactions of compound **3** in the glassy state [15]. Compounds **4** and **5** did not show ability of glass-formation. Exothermal crystallization signals were observed at 160 and 248, 278 °C for **4** and **5**, respectively.

### 2.3. Theoretical Calculations

Theoretical quantum calculations based on DFT/B3LYP/6-31* using Spartan ’14 package software were carried out to understand photophysical and electrochemical properties of target compounds. The overall optimized geometries and distribution of highest occupied molecular (HOMO) and lowest unoccupied molecular orbitals (LUMO) for the ground state of compounds **1**–**6** are illustrated in Figure 2. All compounds along the series adopt highly twisted non-planar configurations, preferable for the active rotations of phenyl rings in the solutions, and are restricted in the solid state. The HOMO of all the compounds **1–6** was found to be similar and localized mainly on the acridan unit and slightly on the connected phenyl ring. Meanwhile the LUMO mainly located on twisted tri- or tetraphenyl ethylene units. Such a HOMO–LUMO separation can be explain by close to perpendicular molecular geometries with dihedral angles between acridan electron-donating fragment and tri- or tetraphenyl frament of 85–88°. Moreover, HOMO–LUMO separation imparts the luminogens with intramolecular charge-transfer characteristics confirmed by experimental photophysical measurements described below (Figure 2). The theoretically calculated HOMO energies were found to be in the range from 4.7 eV to 4.9 eV while the LUMO energies varied from 1.3 eV up to 2.1 eV (Table 2). Due to the presence of additional electron-donating di-*tert*-butyl groups the HOMO energies of compounds 4–6 were found to be slightly lower in respect to the corresponding non-substituted acridan-based compounds **1**–**3**. The same tendency was observed for the LUMO energies of compounds. The theoretically calculated energy gaps varied in the range of 2.7–3.6 eV with the highest values for the compounds with tri- and tetraphenyl ethene fragments (3.3 eV and 3.4 eV) (Table 2). Due to the presence of electron-withdrawing cyano-group compounds **2** and **5** exhibited lower energy gap values of 2.8 and 2.7 eV, respectively (Table 2).

### 2.4. Electrochemical and Photoelectrical Properties.

Electrochemical properties of the compounds were studied by cyclic voltammetry (CV). The ionization potential values (IP_CV_) were determined from the values of the first onset oxidation potential with respect to ferrocene (Figure 3a and Appendix A). The IP_CV_ values of synthesized compounds ranged from 5.40 to 5.63 eV (Table 2). Ionization potentials of vacuum deposited films (IP_EP_) of compounds **1**–**6** were also determined by electron photoemission method in air (Figure 3b). The IP_EP_ values of **1**–**6** ranged from 5.43 to 5.81 eV. The ionization potential values of *tert*-butyl substituted compounds **4**–**6** were found to be lower than those of compounds containing no *tert*-butyl groups (**1**–**3**) due to the slight donating effect of these groups. IP values of compounds **2** and **5** are higher than those of the rest compounds due to the accepting properties of the cyano-group.

### 2.5. Photophysical Properties 

To investigate electronic structures of differently substituted compounds **1**–**6** in the ground state, absorption spectra of their dilute toluene and THF solutions were recorded (Figure 4). The wavelengths of absorption spectra maxima of the solutions of compounds **1**–**6** were not particularly sensitive to the solvents used and were observed at ca. 290 nm. The position of this UV band is close to that of acridan (Figure 4a). This observation shows that the low energy bands of compounds **1**–**6** can be mainly attributed to the local acridan transitions. The shoulders (maxima) at ca. 310–320 nm are attributed to the influence of tetra-/triphenylethenyl moieties. Weak lower-energy absorption bands in the range of 350–450 nm are apparently related to intramolecular charge transfer (ICT) between acridan and tetra-/triphenylethenyl moieties (Figure 4a, inset). This observation can apparently be explained by HOMO–LUMO separation of the compounds due to their twisted molecular structures (Figure 2). Compounds **2** and **5** displayed the most red-shifted ICT bands due to the presence of relatively strong electron-acceptors, i.e., cyano groups. UV absorption spectra of vacuum-deposited films of **1**–**6** replicated the spectra of the corresponding solutions well (Figure 4b). Slightly shifted low energy edges of UV absorption spectra of vacuum-deposited films in comparison to those of solutions can be explained either by stronger ICT of **1**–**6** in solid-state or by aggregation effects.

Because of ICT, PL spectra of compounds **1**–**6** were found to be sensitive to polarity of the media. Thus, PL spectra of THF solutions of compounds **2** and **5** were significantly red-shifted in comparison to their toluene solutions (Figure 4c). Weak red-shifts were also observed for compounds **1**, **3**, **4**, and **6**, containing no cyano groups, induced by the molecular twisting. The introduction of additional phenyl group into phenylethenyl moieties resulted in red-shifts of the fluorescence spectra (cf. the spectra of **3** and **6** with those of **1** and **4**). PL spectra of solid films of compounds **1**–**6** were found to be similar to PL spectra of the corresponding toluene solutions apparently because low dielectric constants of the solid samples (close to that of toluene). Blue-shifted emission of the solid films of compounds **1**–**6** with respect to that of THF solutions may be explained by the influence of relaxation of local exited (LE) states. This assumption is in agreement with double exponential photoluminescence decays of the films of **1**–**6** that can be related to overlapping of relaxation of LE and ICT states (Figure 4d, Appendix A). Solid films of compounds **1**, **3**, **4**, and **6** emitted in the blue region with PL spectra peaked at 458–482 nm (Figure 4b). However, yellowish-green emission was observed for the films of compounds **2** and **5** due to the presence of cyano groups. PL decays of **1**–**6** were observed in ns-range that prove simple fluorescent nature of emission (Figure 4d). Faster fluorescence transients mostly with mono exponential fitting were observed for the solutions of compounds **1**–**6** relative to those of solid samples.

Fluorescence quantum yields (PLQY) of dilute solutions in toluene and of non-doped and doped films of the compounds **1**–**6** are given in Table 3. The films of all the studied compounds exhibited considerably higher PLQY than the corresponding dilute solutions. This observation indicates aggregation induced emission (AIE) for compounds **2**, **3**, **5**, and **6**, with practically absent emission of solutions, and aggregation induce emission enhancement (AIEE) for compounds **1** and **4** with relatively strong emission of the solution in toluene (PLQYs of 32 and 39%). Since non-radiative rates of toluene solutions of compounds **1** and **4** were much lower than those of toluene solutions of other compounds (Appendix A), different PLQY values were obtained for compounds 1–6. PLQY values (26–53%) of non-doped films of the compounds were still much below unity apparently because of intermolecular quenching which may be partly overcome by appropriate hosting [32]. Indeed, using 1,3-bis(N-carbazolyl)benzene (mCP) as the host, the doped films with 10% wt. of the guest showed improved PLQY reaching 88% in case of compound **2** doped in mCP due to its lowest non-radiative rate in comparison to that of other compounds (Appendix A). This finding highlights potential of the compounds for the application in OLEDs. 

To investigate AIE/AIEE of **1**–**6**, PL spectra of their dispersions in the THF-water mixtures with various water fractions (f_w_) were recorded (Figure 5a,b, Appendix A). Being insoluble in water, emissive aggregates of **1**–**6** were formed at the certain concentration of water highlighting AIE/AIEE phenomena. Relative dependences of intensities and wavelengths of PL peaks of compounds **1**–**6** versus water fractions are shown in Figure 4c and Figure 4d respectively. With the increase in water fraction in THF-water mixture, emission intensity of the dispersion of compound **1** constantly decreased and PL maximum wavelength red-shifted until the aggregates were formed (Figure 4c,d). These effects were caused by increasing polarity of the THF-water mixtures to which ICT fluorescence is very sensitive. The further increase of f_w_ lead to the increase of emission intensity and blue shifts of PL spectra due to the increasing amount of aggregates. The similar regularities were observed for the compounds **2**–**4**, **6** and were in good agreement with those reported for many other AIE/AIEE compounds [33,34,35]. However, slightly different behavior was observed for compound **5** (Figure 5b). The dispersion of compound **5** showed maximum emission intensity and higher blue shift at f_w_ = 70%. The further increase of f_w_ induced decrease of PL intensity and PL red-shift (Figure 4c,d). Similar observation was previously detected for compounds with AIE/AIEE effects, although the reason for this is not clear yet [36,37,38]. To our opinion, this observation can apparently be explained by structural modification of aggregates, when their sizes were further increased. 

### 2.6. Electroluminescent Properties 

Since compounds **1**–**6** showed high PLQYs in solid-state, they were tested as emitters for non-doped fluorescent OLEDs. Taking into account the values of ionization potentials and electron affinities obtained for vacuum-deposited films of compounds **1**–**6**, their electroluminescent properties were studied using device structure: ITO/MoO_3_ (0.5 nm)/NPB (35 nm)/mCP (7 nm)/ light-emitting layer (20 nm)/TSPO1 (7 nm)/TPBi (30 nm)/LiF (0.5 nm)/Al. Non-doped light-emitting layers of compounds **1**–**6** were used in devices **1N**–**6N**, respectively. The layers of MoO_3_, NPB, mCP, TSPO1, TPBi, and LiF were used as hole-injecting layer, hole transporting layer, exciton blocking layer, hole/exciton blocking layer, electron transporting layer, and electron-injecting layer, respectively. According to an equilibrium energy diagram of the devices that demonstrates absence of big energy barriers for transported charges under applied external voltages (Appendix A), both holes and electrons were effectively injected to light-emitting layers. Light-emitting recombination of the formed excitons occurred within light-emitting layers. Thus is evident from the shapes of electroluminescence (EL) spectra of devices **1N**–**6N** that were very similar to the shapes of PL spectra of vacuum-deposited films **1–6**, respectively (Figure 6a). EL spectra of devices **1N**–**6N** were found to be similar under different applied voltages proving the main contribution of emitters **1**–**6** in electroluminescence (Appendix A). Blue EL with close CIE coordinates (x from 0.15 to 0.17 and y from 0.13 to 0.25) was observed for devices **1N**, **3N**, **4N**, and **6N** based on acridan and tetra-/triphenylethenyl-based emitters **1**, **3**, **4**, and **6**, containing no cyano groups (Table 4). As seen in Appendix A, the device 1N is characterized by the most blue-shifted electroluminescence with CIE color coordinates (0.15, 0.13) in comparison to that of previously published devices based on tetra(tri)phenylethene-substituted carbazole or triphenylamine OLED emitters. Meanwhile, OLEDs with the emitting layers of compounds **2** and **5** demonstrated yellow EL with CIE of (0.34, 0.56) and (0.41, 0.53) for devices **2N** and **5N** similarly to PL of vacuum-deposited films (Figure 3).

Brightness exceeded of 1000 cd/m^2^ for all non-doped devices **1N**–**6N**. It reached maximum value of 4940 cd/m^2^ in case of device **2N**. EL spectrum of this device was closest to the sensitivity of human eye. In addition, the film of emitter **2** was characterized by high PLQY of 53% (Table 3). Turn on voltages of devices **1N**–**6N** were observed in the range of 4.2–5.4 V demonstrating satisfactory charge-injecting and charge-transporting properties of the devices. Maximum external quantum efficiencies (EQE) of devices **1N**–**6N** were roughly proportional to PLQY values of the non-doped films of the corresponding emitters. (Figure 6d, Table 4). These EQEs are close to those of blue/green devices based on carbazole/triphenylamine and tetra-/triphenylethenyl-containing derivatives [22,39]. The highest EQE of 1.59% was obtained for device **2N**. This value is lower than 2.65%, which is theoretical maximum of EQE for device with fluorescent emitter having PLQY of 53%. Theoretical EQE for device **2N** was calculated by formula η_ext_ = γ × ϕ_PL_ × χ × η_out_ using the charge-balance factor γ = 1, the efficiency of exciton production χ = 0.25 (as for fluorescent emitter), the outcoupling efficiency η_out_ = 0.2, and ϕ_PL_ = 0.53 for the film of compound **2**. Apparently, the charge-balance factor of the studied devices is lower than unity. This presumption can be supported by poor charge-transporting properties of the emitters **1**–**6**. We tried to measure charge mobilities in vacuum-deposited films of compounds **1**–**6** by the time of flight (TOF) method, however, the transit times were not observed possibly because of the fast relaxation of charges in their layers (Appendix A). TOF measurements roughly demonstrated the charge transporting “problems” of the non-doped films of compounds **1**–**6**. Therefore, usage of appropriate hosts was essential. The commonly used host mCP was chosen for the fabrication of doped OLEDs exploiting the same device structure as for non-doped devices. In doped devices **1D**–**6D**, light-emitting layers **1**–**6** (10 wt.%) doped in mCP were used. Selection of the host was based not only on its appropriate HOMO/LUMO energy levels but also on high PLQYs of the films of **1**–**6** (10 wt.%) doped in mCP, which ranged from 38 to 88%. Usage of the host allowed the increase of PLQY of compounds **1**–**6** in solid-state, apparently, due to the decrease of intermolecular interactions (restrictions of π–π* stacking) between neighboring molecules. PL spectra of the doped films were slightly blue-shifted in comparison to PL spectra of the corresponding non-doped films. Polarity and aggregation effects (Figure 3 and Appendix A) can explain this observation. EL spectra of doped devices **1D**–**6D** were in good agreement with PL spectra of light-emitting layers **1**–**6**: mCP (Appendix A). The shapes of EL spectra of the doped devices were the practically same under different external voltages (Appendix A). CIE coordinates of the doped devices **1D**–**6D** were slightly shifted to deeper blue region in comparison to those of non-doped devices **1N**–**6N** (Table 4).

As it was expected, maximum EQEs of all the doped devices (1D–6D) were improved in comparison to those of non-doped ones mainly due to the incensed PLQYs of the emitters dispersed in host and due to the satisfactory charge-transporting properties of mCP (Table 4) [40]. The highest maximum EQE of 2.32% was also obtained for device **2D** based on compound **2**, which showed the highest PLQY of 88% when dispersed in mCP. However, charge-injection properties of the doped devices were not improved. This is evident taking into account the higher turn on voltages of devices **1D**–**6D** compared to those of devices **1N**–**6N**. This observation can be attributed to induced energy barrier in the device structure by relatively deep HOMO of mCP. Nevertheless, it is demonstrated that compounds **1**–**6** can be used as fluorescent emitters in doped OLEDs. When appropriate host exhibiting thermally activated delayed fluorescence (TADF) is available, it is worth testing compounds **1**–**6** as fluorescent emitters in three component systems of **1**–**6**: TADF host:host for increasing efficiencies of **1**–**6**-based OLEDs, keeping in mind that exciton production probability χ = 1 for TADF based systems [10]. In addition, compounds **1**–**6** are potential candidates for sensing applications since they exhibit different emission intensities in liquids and solids.

## 3. Materials

All the required chemicals, i.e., 2-(4-bromophenyl)-1,1-diphenylethenylene, 2-(4-bromophenyl)-1,2,2-triphenylethenylene, solution of tri-*tert*-butylphosphine in toluene (1.00 M), sodium *tert*-butoxide, and palladium acetate were obtained from Sigma-Aldrich and used as received. 9,9-Dimethylacridine and 2,7-di-*tert*-butyl-9,9-dimethylacridine were purchased from Center for Physical Sciences and Technology, Lithuania. 2-(4-Bromophenyl)-2-cyano-1,1-diphenylethylene (Mp = 158–160 °C) was obtained according to the previously described procedure [41].

*10-(4-(2,2-Diphenylethenyl)phenyl)-9,9-dimethylacridine* (**1**). 9,9-Dimethylacridine (0.70 g, 4.78 mmol) and 2-(4-bromophenyl)-1,1-diphenylethylene (1.34 g, 5.74 mmol) were dissolved in anhydrous toluene (10 ml) under Ar. Sodium *tert*-butoxide (0.64 g, 9.56 mmol), palladium acetate (0.02 g, 0.09 mmol), and a solution of tri-*tert*-butylphosphine in toluene (1.00 M, 0.02 ml, 0.09 mmol) were added to the solution and the reaction mixture was refluxed for 12 h. When the reaction was finished (TLC control), the mixture was cooled down to the room temperature and extracted with ethyl acetate. The organic extract was washed with water and dried (Na_2_SO_4_). Then, the solvent was evaporated under vacuum. The product was purified by silica gel column chromatography using hexane as an eluent. White crystals were obtained after recrystallization from hexane with the yield of 72% (1.11 g). Mp = 188–191 °C. MS (ES^+^), *m/z* = 463 [M]^+^. ^1^H NMR (400 MHz, CDCl_3_) δ (ppm): 1.57 (s, 6H), 6.19 (d, *J* = 8.1 Hz, 2H), 6.82 (t, *J* = 7.3 Hz, 2H), 6.88 (t, *J* = 7.6 Hz, 2H), 6.98–7.02 (m, 3H), 7.17 (d, *J* = 8.3 Hz, 2H), 7.19–7.22 (m, 2H), 7.23–7.32 (m, 8H), 7.34 (d, *J* = 6.8 Hz, 2H). ^13^C NMR (100 MHz, CDCl_3_) δ (ppm): 31.1, 35.9, 114.1, 120.5, 125.1, 126.3, 127.2, 127.6, 127.7, 127.8, 128.3, 128.8, 130.1, 130.3, 130.8, 131.9, 137.4, 139.4, 140.1, 140.9, 143.2, 143.7. IR ν_max_ (KBr): 3054, 3028 (C–H, Ar); 2958 (C–H); 1587 (C_6_H_5_-); 1471, 1446 (–CH_3_); 1321 (Ph–CH_3_); 1267 (C–N–, Ar); 923, 751 (C=C–H); 698 (CH=CH).

*10-(4-(1-Cyano-2,2-diphenylethenyl)phenyl)-9,9-dimethylacridine* (**2**). Compound **2** was prepared by a similar procedure to compound **1**, using 9,9-dimethylacridine (0.40 g, 4.78 mmol), 2-(4-bromophenyl)-2-cyano-1,1-diphenylethylene (0.83 g, 5.74 mmol), sodium *tret*-butoxide (0.37 g, 9.56 mmol), palladium acetate (0.01 g, 0.09 mmol), and a solution of tri-*tert*-butylphosphine in toluene (1.00 M, 0.01 ml, 0.09 mmol). Light yellow crystals were obtained after recrystallization from hexane with the yield of 83 % (0.77 g). Mp = 219–222 °C. MS (ES^+^), *m/z* = 488 [M]^+^. ^1^H NMR (400 MHz, CDCl_3_) δ (ppm): 1.59 (s, 6H), 6.14 (d, *J* = 7.9 Hz, 2H), 6.86 (t, *J* = 7.3 Hz, 2H), 6.89–6.95 (m, 2H), 7.01 (d, *J* = 7.2 Hz, 2H), 7.11 (d, *J* = 8.2 Hz, 2H), 7.14–7.23 (m, 2H), 7.32–7.45 (m, 10H). ^13^C NMR (100 MHz, CDCl_3_) δ (ppm): 31.2, 35.9, 110.7, 113.9, 119.9, 120.8, 125.3, 126.4, 128.3, 128.6, 129.4, 130.0, 130.1, 130.2, 130.9, 131.5, 132.3, 135.1, 138.9, 139.9, 140.6, 141.1, 158.9. IR ν_max_ (KBr): 3050, 3032 (C–H, Ar); 2975, 2957 (C–H); 2208 (–C≡N); 1588, 1506 (C_6_H_5_–); 1472, 1445 (–CH_3_); 1324 (Ph–CH_3_); 1270 (C–N–, Ar); 1110 (–C–N–); 922, 748 (C=C–H); 704 (CH=CH).

*10-(4-(1,2,2-Triphenylethenyl)phenyl)-9,9-dimethylacridine* (**3**). Compound **3** was prepared by the similar procedure as compound **1**, using 9,9-dimethylacridine (0.40 g, 4.78 mmol), 2-(4-bromophenyl)-1,1,2-triphenylethylene (0.94 g, 5.74 mmol), sodium *tret*-butoxide (0.37 g, 9.56 mmol), palladium acetate (0.01 g, 0.09 mmol), and a solution of tri-*tert*-butylphosphine in toluene (1.00 M, 0.01 ml, 0.09 mmol). White crystals were obtained after recrystallization from hexane with the yield of 68% (0.70 g). Mp = 245–248 °C. MS (ES^+^), *m/z* = 539 [M]^+^. ^1^H NMR (400 MHz, CDCl_3_) δ (ppm): 1.66 (s, 6H), 6.25 (d, *J* = 8.1 Hz, 2H), 6.93 (t, *J* = 7.4 Hz, 2H), 7.01 (t, *J* = 7.7 Hz, 2H), 7.07 (d, *J* = 8.2 Hz, 2H), 7.10–7.20 (m, 15H), 7.30 (d, *J* = 8.2 Hz, 2H), 7.45 (d, *J* = 7.6 Hz, 2H). ^13^C NMR (100 MHz, CDCl_3_) δ (ppm): 31.3, 36.1, 114.1, 120.6, 125.3, 126.4, 126.8, 126.9, 127.8, 127.9, 128.0, 130.1, 130.6, 131.4, 131.6, 133.8, 139.2, 140.5, 140.9, 142.1, 143.1, 143.3, 143.7, 144.1. IR ν_max_ (KBr): 3052, 3024 (C–H, Ar); 2948 (C–H); 1590, 1508 (C_6_H_5_–); 1475, 1441 (–CH_3_); 1324 (Ph–CH_3_); 1270 (C–N–, Ar); 921, 748 (C=C–H); 694 (CH=CH).

*2,7-Di-*tert*-butyl-10-(4-(2,2-diphenylethenyl)phenyl)-9,9-dimethylacridine* (**4**). Compound **4** was prepared by the similar procedure as compound **1**, using 2,7-di-*tert*-butyl-9,9-dimethylacridine (0.70 g, 3.11 mmol), 2-(4-bromophenyl)-1,1-diphenylethylene (0.88 g, 3.74 mmol), sodium *tret*-butoxide (0.42 g, 6.22 mmol), palladium acetate (0.01 g, 0.06 mmol), and a solution of tri-*tert*-butylphosphine in toluene (1.00 M, 0.01 ml, 0.06 mmol). Light yellow crystals were obtained after recrystallization from hexane with the yield of 89% (1.12 g). Mp = 192–196 °C. MS (ES^+^), *m/z* = 576 [M]^+^. ^1^H NMR (400 MHz, CDCl_3_) δ (ppm): 1.22 (s, 18H), 1.60 (s, 6H), 6.02–6.16 (m, 2H), 6.89 (d, *J* = 8.0 Hz, 2H), 6.93–7.03 (m, 3H), 7.10–7.16 (m, 2H), 7.17–7.21 (m, 2H), 7.22–7.32 (m, 8H), 7.36 (s, 2H). ^13^C NMR (100 MHz, CDCl_3_) δ (ppm): 29.8, 31.6, 34.2, 36.4, 113.3, 122.1, 123.0, 127.4, 127.6, 127.7, 127.8, 128.3, 128.8, 130.4, 130.9, 131.9, 137.2, 140.1, 143.3, 143.5. IR ν_max_ (KBr): 3081, 3050 (C–H, Ar); 2949, 2901 (C–H); 1603, 1506 (C_6_H_5_–); 1490 (–CH_3_); 1410, 1361 ((CH_3_)_3_C–); 1330 (Ph–CH_3_); 1265 (C–N–, Ar); 890, 817, 763 (C=C–H); 696 (CH=CH).

*2,7-Di-*tert*-butyl-10-(4-(1-cyano-2,2-diphenylethenyl)phenyl)-9,9-dimethylacridine* (**5**). Compound **5** was prepared by the similar procedure to compound **1**, using 2,7-di-*tert*-butyl-9,9-dimethylacridine (0.40 g, 2.85 mmol), 2-(4-bromophenyl)-2-cyano-1,1-diphenylethylene (0.49 g, 3.41 mmol), sodium *tert*-butoxide (0.22 g, 5.69 mmol), palladium acetate (0.01 g, 0.06 mmol), and a solution of tri-*tert*-butylphosphine in toluene (1.00 M, 0.01 ml, 0.06 mmol). The product was purified by silica gel column chromatography using an eluent mixture of THF and hexane in the volume ratio of 1:50. Yellow crystals were obtained after recrystallization from the eluent mixture of solvents with the yield of 85% (0.64 g). Mp = 272–275 °C. MS (ES^+^), *m/z* = 600 [M]^+^. ^1^H NMR (400 MHz, CDCl_3_) δ (ppm): 1.24 (s, 18H), 1.61 (s, 6H), 5.98–6.13 (m, 2H), 6.93 (d, *J* = 8.0 Hz, 2H), 6.99 (d, *J* = 9.8 Hz, 2H), 7.10 (d, *J* = 7.2 Hz, 2H), 7.14–7.22 (m, 4H), 7.34–7.47 (m, 8H). ^13^C NMR (100 MHz, CDCl_3_) δ (ppm): 29.9, 31.7, 34.2, 36.4, 110.8, 113.6, 120.0, 123.1, 128.3, 128.6, 129.4, 130.0, 130.2, 130.9, 131.7, 132.2, 134.8, 138.9, 140.1, 158.8. IR ν_max_ (KBr): 3054, 3037 (C–H, Ar); 2957, 2901 (C–H); 2206 (–C≡N); 1639 (C=C, Ar); 1601, 1506 (C_6_H_5_–); 1489 (–CH_3_); 1410, 1361 ((CH_3_)_3_C–); 1328 (Ph–CH_3_); 1267 (C–N–, Ar); 887, 809, 745 (C=C–H); 699 (CH=CH).

*2,7-Di-*tert*-butyl-10-(4-(1,2,2-triphenylethenyl)phenyl)-9,9-dimethylacridine* (**6**). Compound **6** was prepared by the similar procedure as compound **1**, using 2,7-di-*tert*-butyl-9,9-dimethylacridine (0.40 g, 3.11 mmol), 1-(4-bromophenyl)-1,2,2-triphenylethylene (0.61 g, 3.73 mmol), sodium *tert*-butoxide (0.24 g, 6.22 mmol), palladium acetate (0.01 g, 0.06 mmol), and a solution of tri-*tert*-butylphosphine in toluene (1.00 M, 0.01 ml, 0.06 mmol). White crystals were obtained after recrystallization from hexane with the yield of 77% (0.62 g). Mp = 261–263 °C. MS (ES^+^), *m/z* = 652 [M]^+^. ^1^H NMR (400 MHz, CDCl_3_) δ (ppm): 1.24 (s, 18H), 1.60 (s, 6H), 5.98–6.13 (m, 2H), 6.92 (d, *J* = 8.4 Hz, 2H), 6.96 (d, *J* = 8.0 Hz, 2H), 7.00–7.11 (m, 15H), 7.15–7.19 (m, 2H), 7.37 (s, 2H). ^13^C NMR (100 MHz, CDCl_3_) δ (ppm): 25.1, 31.6, 34.2, 36.7, 113.3, 122.3, 122.9, 126.6, 126.7, 126.8, 127.6, 127.8, 127.9, 129.5, 130.6, 131.4, 131.5, 133.6, 139.5, 140.5, 141.9, 143.1, 143.3, 143.7, 143.8. IR ν_max_ (KBr): 3054, 3039 (C–H, Ar); 2953, 2896 (C–H); 1603, 1509 (C_6_H_5_–); 1493, 1443 (–CH_3_); 1412, 1362 ((CH_3_)_3_C–); 1336 (Ph–CH_3_); 1267 (C–N–, Ar); 890, 808, 744 (C=C–H); 696 (CH=CH).

Devices were fabricated using the synthesized materials as AIE/AIEE emitters and commercially available molybdenum trioxide (MoO_3_), N,N′-di(1-naphthyl)-N,N′-diphenyl-(1,1′-biphenyl)-4,4′-diamine (NPB), 1,3-bis(N-carbazolyl)benzene (mCP), diphenyl[4-(triphenylsilyl)phenyl]phosphine oxide (TSPO1), 2,2′,2″-(1,3,5-benzinetriyl)-tris(1-phenyl-1-H-benzimidazole) (TPBi), fluorolithium (LiF) as additional functional layers.

## 4. Conclusions

We synthesized and characterized six tetra-/triphenylethene-substituted 9,9-dimethylacridine derivatives, which exhibited aggregation induced emission (enhancement) allowing achieving high fluorescence quantum yields of 26–53% for non-doped and 52–88% for doped films. Cyano-substitution resulted in increase of ionization potentials of the derivatives and lead to significant red-shifts of emission. Non-doped and doped films of four tetra-/triphenylethene-substituted 9,9-dimethylacridine derivatives were characterized by blue (451–481 nm) and deep-blue (438–445 nm) fluorescence in respectively. Utilizing the synthesized compounds in organic light-emitting diodes, deep-blue electroluminesce with chromaticity coordinates of (0.16, 0.10), close to the blue color standard (0.14, 0.08) of the National Television System Committee, were obtained. The devices exhibited external quantum efficiencies up to 2.3%.

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
