# Peer review of "Towards Blue AIE/AIEE: Synthesis and Applications in OLEDs of Tetra-/Triphenylethenyl Substituted 9,9-Dimethylacridine Derivatives"

_molecules, 2020, doi:10.3390/molecules25030445_

Round 1
Reviewer 1 Report
Dear Authors,
The manuscript clearly describes the synthesis and application of six dimethylacridine derivatives as AIE/AIEE blue emitters in Organic Light-Emitting Diodes. The characterisation of the materials is detailed and well described and OLED fabrication method is experimentally sound and the results explained in detail. Good design of experiment, scientific soundness and quality of presentation makes this an enjoyable and easy to read manuscript.
I recommend publication after making the following revisions that would improve the manuscript further.
1) The explanation of melting points for compounds 4 and 6 needs some consideration and clarification. Line 104-110 makes the statement that tert-butyl analogues have a higher melting point due to rigidity. However, you would expect the additional steric hindrance created by the bulky alkyl chains to create less rigidity.
2) Add experimental conditions and reagents to scheme 1.
3) Line 202-203 explains the reason for compound 5 having a fw = 70% as a statement of fact without supporting evidence. I would suggest referencing were this effect has been seen previously.
4) Add the NMR spectra of compounds 1-6 in the Supporting Information.
I enjoyed reading your manuscript and commend you for the quality of presentation, design of experiment and scientific soundness.
Author Response
Reviewer 1
The manuscript clearly describes the synthesis and application of six dimethylacridine derivatives as AIE/AIEE blue emitters in Organic Light-Emitting Diodes. The characterisation of the materials is detailed and well described and OLED fabrication method is experimentally sound and the results explained in detail. Good design of experiment, scientific soundness and quality of presentation makes this an enjoyable and easy to read manuscript.
I recommend publication after making the following revisions that would improve the manuscript further.
Response of the authors.
We appreciate that the Reviewer #1 recognizes the novelty and impact of this work. In addition, we thank Reviewer #1 for the comments, which allow to improve the quality of our manuscript. We hope that the changes we made in response to the suggestions will further support our conclusions and will help to clarify the novelty and impact of our work.
1) The explanation of melting points for compounds 4 and 6 needs some consideration and clarification. Line 104-110 makes the statement that tert-butyl analogues have a higher melting point due to rigidity. However, you would expect the additional steric hindrance created by the bulky alkyl chains to create less rigidity.
Response of the authors.
The explanation of melting points for compounds 4 and 6 was reconsidered as following:
Introduction of larger substituents such as cyano- group (compounds 2, 5) and phenyl (compounds 3, 6) fragment instead of hydrogen (compounds 1, 4) in the ethylene unit highly increased the Tm and Td values of compounds (Table 1). Moreover, the introduction of bulky tert-butyl groups apparently affects the crystal packing between adjacent molecules and conformational arrangements of acridan units that resulted in increased Tm values for compounds 4-6 in respect to non-substituted compounds 1-3 [29, 30]. The attachment of tert-butyl substituents seems to make structures even more bulky and rigid which presumably leads to the enhanced Tm of compounds 4–6 as well as to their tendency to crystallize from liquid phase [30].
2) Add experimental conditions and reagents to scheme 1.
Response of the authors.
Scheme 1 had been edited and experimental conditions and reagents were added to its caption.
3) Line 202-203 explains the reason for compound 5 having a fw = 70% as a statement of fact without supporting evidence. I would suggest referencing were this effect has been seen previously.
Response of the authors.
Indeed, similar observation was previously detected for compounds with AIE/AIEE effects, while its reasons is not clear yet [36-38]. To our opinion, this observation can apparently be explained by structural modification of aggregates, when their sizes were further increased leading to molecular geometry changes resulting in the emission red-shift.
Han, T.; Hong, Y.; Xie, N.; Chen, S.; Zhao, N.; Zhao, E.; Lam, J.W.Y.; Sung, H.H.Y.; Dong, Y.; Tong, B.; Tang. B.Z. Defect-sensitive crystals based on diaminomaleonitrilefunctionalized Schiff base with aggregation-enhanced emission. J. Mater. Chem. C, 2013, 1, 7314. Zhang, X.; Chi, Z.; Xu, B.; Chen, C.; Zhou, X.; Zhang, Y.; Liu, S.; Xu. J. End-group effects of piezofluorochromic aggregation-induced enhanced emission compounds containing distyrylanthracene. J. Mater. Chem., 2012, 22, 18505. Yang, Z.; Chi, Z.; Xu, B.; Li, H.; Zhang, X.; Li, X.; Liu, S.; Zhang. Y.; Xu. J. High-Tg carbazole derivatives as a new class of aggregation-induced emission enhancement materials. J. Mater. Chem., 2010, 20, 7352.
4) Add the NMR spectra of compounds 1-6 in the Supporting Information.
I enjoyed reading your manuscript and commend you for the quality of presentation, design of experiment and scientific soundness.
Response of the authors.
1H NMR spectra of synthesized compounds were included to Supporting information.
Reviewer 2 Report
This paper reports interesting results on AIE/AIEE active materials which can be applicable for non-doped/doped OLED emitting layers. Even though the material design is a kind of typical AIE type approach, the authors systematically well studied the structure-property relationships. This report has broad appeals to the general readership for this journal. However, I would like to suggest the authors include or correct some points in order to increase the completeness of this research.
In page 5, the authors explained the ICT bands and HOMO-LUMO charge separation. This discussion can be improved when it combines with DFT calculation and the explanation of HOMO-LUMO orbital distributions. The PL decays in toluene solutions (Figure 3d) are not matched with those in Figure S3a. The compound numbers in Figure S3a should be changed. In the PLQYs of toluene solutions, compound 1 and 4 have significantly large PLQYs compared with other compounds. It seems that additional CN/phenyl attachment into triphenylethenyl unit affects molecular geometry thus bring increasement of non-radiative decay (Knr). It is needed to compare the molecular geometry and twist angle between 1, 4 compounds and the other compounds. It is also helpful to explain this significant different PLQYs of the compounds in toluene solutions, if the authors can include Kr and Knr values which can be calculated from PLQY and fluorescence decay time and combine with the results of molecular geometry calculation. Some figure and compound numbers are nor correct and a figure is missing.In page 7 line 2, compounds 1 and 3 should be changed to compounds 1 and 4.
In page 7 line 10, S4 should be changed to S5.
In page 8 line 10, S5 should be changed to S6.
In page 8 line 14-16, the authors discussed about EL spectra under different applied voltages. But the figure is missing. The figure should be included in the supporting information.
Author Response
Reviewer 2
This paper reports interesting results on AIE/AIEE active materials which can be applicable for non-doped/doped OLED emitting layers. Even though the material design is a kind of typical AIE type approach, the authors systematically well studied the structure-property relationships. This report has broad appeals to the general readership for this journal. However, I would like to suggest the authors include or correct some points in order to increase the completeness of this research.
Response of the authors.
We appreciate that the Reviewer #2 recognizes the novelty and impact of this work. In addition, we thank to Reviewer #2 for the comments that allow to improve the quality of our manuscript. We hope that the changes we made in response to the suggestions will further support our conclusions and will help to more clearly disclose the novelty and impact of our work.
In page 5, the authors explained the ICT bands and HOMO-LUMO charge separation. This discussion can be improved when it combines with DFT calculation and the explanation of HOMO-LUMO orbital distributions.
Response of the authors.
To support the statements related to the ICT bands and HOMO-LUMO charge separation, theoretical quantum calculations based on DFT/B3LYP/6-31* using Spartan ’14 package software were performed and the manuscript was accordingly updated by following paragraph:
2.3. Theoretical calculations.
Theoretical quantum calculations based on DFT/B3LYP/6-31* using Spartan ’14 package software were carried out to understand photophysical and electrochemical properties of target compounds. The overall optimized geometries and distribution of highest occupied molecular (HOMO) and lowest unoccupied molecular orbitals (LUMO) for the ground state of compounds 1-6 are illustrated in Figure 2. All compounds along the series adopt highly twisted non-planar configurations, preferable for the active rotations of phenyl rings in the solutions and are restricted in the solid state. The HOMO of all the compounds 1-6 was found to be similar and localized mainly on the acridan unit and slightly on the connected phenyl ring. Meanwhile the LUMO mainly located on twisted tri- or tetraphenyl ethylene units. Such a HOMO-LUMO separation can be explain by close to perpendicular molecular geometries with dihedral angles between acridan electron-donating fragment and tri- or tetraphenyl frament of 85 – 88°. Moreover, HOMO-LUMO separation imparts the luminogens with intramolecular charge-transfer characteristics confirmed by experimental photophysical measurements described below (Figure 2). The theoretically calculated HOMO energies were found to be in the range from 4.7 eV up to 4.9 eV while the LUMO energies varied from 1.3 eV up to 2.1 eV (Table 2). Due to the presence of additional electron-donating di-tert-butyl groups the HOMO energies of compounds 4-6 were found to be slightly lower in respect to the corresponding non-substituted acridan-based compounds 1-3. The same tendency was observed for the LUMO energies of compounds. The theoretically calculated energy gaps varied in the range of 2.7 – 3.6 eV with the highest values for the compounds with tri- and tetraphenyl ethene fragments (3.3 eV and 3.4 eV). Due to the presence of electron-withdrawing cyano-group compounds 2 and 5 exhibited lower energy gap values of 2.8 and 2.7 eV, respectively (Table 2).
The PL decays in toluene solutions (Figure 3d) are not matched with those in Figure S3a. The compound numbers in Figure S3a should be changed.
Response of the authors.
The Figure S3a was corrected.
In the PLQYs of toluene solutions, compound 1 and 4 have significantly large PLQYs compared with other compounds. It seems that additional CN/phenyl attachment into triphenylethenyl unit affects molecular geometry thus bring increasement of non-radiative decay (Knr). It is needed to compare the molecular geometry and twist angle between 1, 4 compounds and the other compounds. It is also helpful to explain this significant different PLQYs of the compounds in toluene solutions, if the authors can include Kr and Knr values which can be calculated from PLQY and fluorescence decay time and combine with the results of molecular geometry calculation.
Response of the authors.
Indeed, slightly higher dihedral angels between the acridan electron-donating fragment and tri- or tetraphenyl fragment were obtained for 1, 4 compounds (Figure 2).
Additionally, the radiative (kr) and non-radiative (knr) rate constants were calculated and collected in Table S1.
As it was expected, non-radiative rates of toluene solutions of compounds 1 and 4 were much lower than toluene solutions of other compounds (Table S1). Since non-radiative rates of toluene solutions of compounds 1 and 4 were much lower than those of toluene solutions of other compounds (Table S1), different PLQY values were obtained for compounds 1-6. While, relatively similar radiative rates were obtained for doped films 1-6 due to the AIE/AIEE effects. The doped films with 10% wt. of the guest showed improved PLQY reaching 88 % in case of compound 2 doped in mCP due to its lowest non-radiative rate in comparison to that of other compounds (Table S1).
Some figure and compound numbers are nor correct and a figure is missing.
In page 7 line 2, compounds 1 and 3 should be changed to compounds 1 and 4.
In page 7 line 10, S4 should be changed to S5.
In page 8 line 10, S5 should be changed to S6.
In page 8 line 14-16, the authors discussed about EL spectra under different applied voltages. But the figure is missing. The figure should be included in the supporting information.
Response of the authors.
The above typo errors were corrected. Figure S7 with EL spectra under different applied voltages of the fabricated devices was included to the supporting information.

Reviewer 3 Report
This paper describes synthesis and physical characteristics of tetra-/triphenylethyl substituted 9,9-dimethylacridine derivatives aiming at an improved performance of organic light-emitting diodes. The Authors designed new derivatives according to previous findings. The obtained data showed an improved performance which is expected by the Author with new findings such as substitution effect. However, it is better to add some explanation how much performance has been improved from the previous data more specifically. This additional explanation might be help to lead the Readers understand.
Therefore, this paper should be published in this journal after this additional explanation
Author Response
Reviewer 3
This paper describes synthesis and physical characteristics of tetra-/triphenylethyl substituted 9,9-dimethylacridine derivatives aiming at an improved performance of organic light-emitting diodes. The Authors designed new derivatives according to previous findings. The obtained data showed an improved performance which is expected by the Author with new findings such as substitution effect. However, it is better to add some explanation how much performance has been improved from the previous data more specifically. This additional explanation might be help to lead the Readers understand.
Therefore, this paper should be published in this journal after this additional explanation
Response of the authors.
We appreciate that Reviewer #3 recognizes the novelty and impact of this work. In addition, we thank Reviewer #3 for the comments which allow to improve of the quality of our manuscript. We hope that the changes we made in response to the suggestions will further support our conclusions and will help to more clearly disclose the novelty and impact of our work.
We compared performances of non-doped devices based on nitrogen-containing heteroaromatic carbazole, triphenylamine and acridan moieties substituted by tetraphenylethene and/or triphenylethene units (Table S2). As it is seen from the Table S2, the device 1N is characterized by the most blue-shifted electroluminescence with CIE color coordinates (0.15, 0.13) in comparison to that of previously published devices based on tetra(tri)phenylethene-substituted carbazole or triphenylamine OLED emitters. While, electroluminescence of doped device 1D is even more blue-shifted (Figure 5a, Table 4).
